# Characterization of Hantavirus N Protein Intracellular Dynamics and Localization

**DOI:** 10.3390/v14030457

**Published:** 2022-02-23

**Authors:** Robert-William Welke, Hannah Sabeth Sperber, Ronny Bergmann, Amit Koikkarah, Laura Menke, Christian Sieben, Detlev H. Krüger, Salvatore Chiantia, Andreas Herrmann, Roland Schwarzer

**Affiliations:** 1Department of Molecular Biophysics, Humboldt University, 10115 Berlin, Germany; robert-william.welke@gmx.de (R.-W.W.); ronny.bergmann91@gmail.com (R.B.); andreas.herrmann@rz.hu-berlin.de (A.H.); 2Institute for Translational HIV Research, University Hospital Essen, 45147 Essen, Germany; hannahsperber.7@googlemail.com; 3Institute of Biochemistry and Biology, University of Potsdam, 14476 Potsdam, Germany; amitkoikkarah93@gmail.com (A.K.); chiantia@uni-potsdam.de (S.C.); 4Nanoscale Infection Biology Group, Department of Cell Biology, Helmholtz Centre for Infection Research, 38124 Braunschweig, Germany; laura.menke@helmholtz-hzi.de (L.M.); christian.sieben@helmholtz-hzi.de (C.S.); 5Institute for Genetics, Technische Universität Braunschweig, 38106 Braunschweig, Germany; 6Institut für Virologie, Charité–Universitätsmedizin Berlin, Gliedkörperschaft der Freien Universität Berlin und der Humboldt-Universität zu Berlin, 10117 Berlin, Germany; detlev.krueger@charite.de; 7Biophysikalische Chemie, Freie Universität, 14195 Berlin, Germany

**Keywords:** hantavirus, N protein, oligomerization, actin, P-bodies, vimentin, Number and Brightness, Puumalavirus, macromolecular assemblies

## Abstract

Hantaviruses are enveloped viruses that possess a tri-segmented, negative-sense RNA genome. The viral S-segment encodes the multifunctional nucleocapsid protein (N), which is involved in genome packaging, intracellular protein transport, immunoregulation, and several other crucial processes during hantavirus infection. In this study, we generated fluorescently tagged N protein constructs derived from Puumalavirus (PUUV), the dominant hantavirus species in Central, Northern, and Eastern Europe. We comprehensively characterized this protein in the rodent cell line CHO-K1, monitoring the dynamics of N protein complex formation and investigating co-localization with host proteins as well as the viral glycoproteins Gc and Gn. We observed formation of large, fibrillar PUUV N protein aggregates, rapidly coalescing from early punctate and spike-like assemblies. Moreover, we found significant spatial correlation of N with vimentin, actin, and P-bodies but not with microtubules. N constructs also co-localized with Gn and Gc albeit not as strongly as the glycoproteins associated with each other. Finally, we assessed oligomerization of N constructs, observing efficient and concentration-dependent multimerization, with complexes comprising more than 10 individual proteins.

## 1. Introduction

Hantaviruses (HV, *Hantaviridae*, order *Bunyavirales*) are a family of emerging viruses causing life-threatening human zoonoses with case fatalities of up to 60% [1,2]. In Europe, the less virulent hantavirus species Puumala (PUUV) causes most reported hantavirus-associated diseases.

Puumala virus particles comprise a lipid envelope and a single-stranded, tri-segmented RNA genome that encodes for five proteins [3]: an RNA dependent RNA polymerase RdRp, the glycoproteins Gn and Gc, the non-structural protein NSs, and the nucleocapsid protein N. Entry of HV particles into their target cell, predominantly of the endothelial lineage [4], is mediated by the viral spike complex, a Gc/Gn heterotetramer. We and others have shown that, after engagement with their receptor, old-world hantaviruses exploit multiple entry routes to get access to the vulnerable host cell cytoplasm [5,6,7]. Subsequent virus replication takes place at the endoplasmic reticulum–Golgi intermediate compartment (ERGIC) and involves N- and RdRp-mediated cap-snatching as a prerequisite of viral translation [8]. It has been proposed that, throughout the course of an infection, viral factories are formed, which involves ER and Golgi membranes, P-bodies, and ribosomes as well as multiple cytoskeleton components [6,8,9]. Virus assembly and budding is then mediated by the three structural proteins, Gc, Gn, and N. The latter binds the viral RNA, thus forming ribonucleoprotein complexes, which eventually recruit the virus genome to the nascent virion [10,11]. Virus particle formation, on the other hand, is believed to be solely controlled by the HV glycoproteins Gc and Gn. Ultimately, Gn and Gc interact with both the vRNA and the associated N proteins [12,13], leading to the formation of mature virions [6]. After budding into the ERGIC, virus particles are then released from the infected cell by mechanisms that are barely understood [6,14]. Throughout the course of an infection cycle, hantaviruses trigger a substantial reorganization of the cytoskeleton and overall structural organization of their host cells [5,15,16]; however, the underlying processes and involved host-pathogen interactions remained cryptic.

This study focuses on one of the key structural proteins of PUUV, the nucleoprotein N. We generated chimeric proteins that harbor fluorescent proteins fused to the N-terminus of N to be able to assess the dynamics of N localization and trafficking in live-cell experiments. We found that YFP-N rapidly clusters in expressing cells, eventually forming macromolecular complexes that can extend through most of the cell body. We also observed preferential co-localization with P-bodies, actin, and vimentin but not tubulin, suggesting selective association with cytoskeleton components. Upon co-expression with other structural PUUV proteins, the glycoproteins Gc and Gn, strong spatial correlation was found in the perinuclear region, likely indicative of nascent virus assembly processes. Finally, using in fluorescence fluctuation spectroscopy experiments, we observed large-scale oligomerization of YFP-N, which did not markedly change when viral glycoproteins were co-expressed.

Our experiments aimed at studying properties and intracellular activities of the PUUV N protein independent of infections and largely in the absence of most other viral proteins and viral genomic RNA. Our goal was to explore inherent properties of N that govern its intracellular localization and processing, which eventually contributes to PUUV particle formation and release. The data therefore shed new light on the intricate interplay between cellular and viral components, which could reveal key vulnerabilities of hantavirus infection cycles.

## 2. Materials and Methods

### 2.1. Generation of Fluorescently Labelled Hantavirus N Protein

Vero E6 cells were infected with the Puumalavirus, strain Sotkamo (V-2969/81), which is an *Orthohantavirus* from the family of the *Hantaviridae*. At day 3 post infection, cells were subjected to mRNA extraction (RNeasy, Qiagen, Hilden, Germany), followed by reverse transcription. Then, cDNA subjected to PCR amplification using primers for PUUV N protein. N protein PCR products were sub-cloned into pmYFP-N1, which harbors a A206K monomeric mutation of the fluorescent tag, using NotI and BsrgI restriction sites. An additional mTurquoise construct was generated by excising mYFP using AgeI and NotI restriction sites and replacing it with a mTurquoise PCR product.

### 2.2. Cell Culture and Transfection

Chinese hamster ovary (CHO-K1) cells and African green monkey kidney epithelial cells (Vero E6 cells) were maintained in Dulbecco’s modified Eagle’s medium (DMEM, PAA Laboratories GmbH, Austria) supplemented with 10% fetal bovine serum, 2 mM L-glutamine, 100 U/mL penicillin, and 100 μg/mL streptomycin (all PAA Laboratories GmbH, Pasching, Austria). Then, 24–48 h prior to imaging experiments, expression plasmids were introduced into pre-plated cells in 35-mm glass-bottom Microwell Dishes (MatTek Corporation, Ashlands, MA, USA) by Turbofect transfection (Thermo Scientific, Waltham, MA, USA) according to the manufacturer’s instructions. All cell lines other than CHO-K1 (Appendix A: A549, HEK293T, MGLU-2-R [17], and MGN-2-R [17]) were seeded in a standard 12-well tissue culture plate (Greiner, Kremsmünster, Austria) on 18-mm glass cover slips (#1.5, Menzel, Thermo Scientific, Waltham, MA, USA). The cells were seeded at a density of 150.000 cells per well and cultivated in DMEM with 10% fetal calf serum (Sigma-Aldrich, Gillingham, UK). After 24 h, the cells were transfected with 1 µg plasmid DNA per well using jetOptimus (Polyplus, Illkirch, France) transfection reagent. The medium was changed after 6 h, and the cells were further incubated for 12–16 h, then fixed, stained, and mounted on microcopy glass slides (ProLong Gold, ThermoFisher, Waltham, MA, USA).

### 2.3. Immunofluorescence Staining

Intracellular immunofluorescence staining of transfected CHO-K1 cells was performed as follows: First, pre-plated cells were washed three times with phosphate-buffered saline with calcium and magnesium (PBS+/+) and fixed for 25 min with 3.7% paraformaldehyde at room temperature. Then, the cells were subjected to three washes with PBS+/+ before permeabilization for 20 min with 0.2% Triton X-100 and 0.2% bovine serum albumin (BSA). Following three more washing steps, cells were incubated with primary antibodies for 1 h at room temperature (RT), washed with PBS+/+ three more times, and incubated with conjugated secondary antibody for 1 h at RT (Table 1). Finally, cells were washed three more times and subjected to microscopy. All cell lines other than CHO-K1 (Appendix A) were fixed with 4% paraformaldehyde for 20 min at room temperature, stained, and mounted on microcopy glass slides (ProLong Gold, Thermo Scientific, Waltham, MA, USA) 12–16 h post transfection. Immunostaining was performed as described using anti-tubulin antibodies and Phalloidin-Alexa647 at a concentration of 0.16 µM.

### 2.4. Fluorescence Microscopy

Confocal spinning disc microscopy (CSD) was used for all antibody stains and long-term exposure experiments. Images were obtained using a Visitron VisiScope scanning-disc confocal laser microscope (Visitron Systems, Puchheim, Germany) utilizing a 60×/1.2 UPlanSApo water or a 100×/1.3 UPlanFLN oil objective (pixel size of 0.13 and 0.2 µm, respectively) and detecting fluorescence with an Andor iXon 888 EMCCD camera (1024 × 1024 pixels, Andor, Belfast, Northern Ireland). The following diode lasers and filter sets were used for fluorescence detection: 488 nm (FITC) with an ET525/50-nm emission filter, 561 nm (Atto550) with an ET600/50-nm emission filter, 640 nm (PCA635P) with an ET700/75-nm emission filter, and 405 nm (DAPI) with an ET460/50-nm emission filter. If not otherwise stated, we acquired z-stacks, the respective images show z-projections, and z-projections were used for quantitative analysis. YFP-N single stains and co-transfection experiments were imaged using an Olympus FluoView 1000 MPE confocal microscope, equipped with 60×/1.2 Water (UPlanSApo) and 60×/1.45 Oil (UPlanSApo) objective, respectively, as well as 405-nm, 440-nm, 561-nm, and 635-nm diode laser and 488-nm and 515-nm Argon lasers with the following filter sets: 80/20, 405-458/515/559/635, 405/488/559/635, 458/515, and 405/458/515. All cell lines other than CHO-K1 (Appendix A) were imaged using a Nikon Ti2 spinning disk confocal microscope.

### 2.5. Image Analysis

Manual and semi-automatic image processing and analysis were performed with ImageJ (https://imagej.nih.gov/ij/, accessed at 22 July 2017). Spot detection (Figure 1) was carried out on maximum-intensity projections of z-stacks using a plug-in developed by Eugene Katrukha (ComDet). Intensity thresholds were defined using no-virus control samples. The following settings were used for analysis: a particle size of 4 pixels and an intensity threshold of 6. The analysis included but did not segment larger particles. Automatic image analysis was performed with CellProfiler [18] using an in-house pipeline (available upon request). Briefly, cells were identified in the DNA staining channel with the “identify primary objects” module. Then, cell bodies were segmented in either actin, microtubule, vimentin, or DIC images with the module “identify secondary objects.” Thereafter, the module “MeasureColocalization” was executed to assess the pixel-by-pixel Pearson correlation coefficient.

### 2.6. Number and Brightness Analysis

Number and Brightness analysis (N&B) was performed as previously described in Petazzi et al. [19]. Briefly, 48 h prior to the experiment, 3 to 6 × 10^5^ cells were plated onto 35-mm glass-bottom dishes (CellVis, Mountain View, CA, USA or MatTek Corp., Ashlands, MA, USA) and transfected with the plasmids of interest. Confocal images were acquired using a Zeiss LSM780 microscope (Carl Zeiss Microscopy GmbH, Jena, Germany). CW Argon laser 488-nm excitation light was focused with an objective into the sample. A Zeiss QUASAR multichannel GaAsP detector was used to collect fluorescence in the 498 to 606 nm range in photon-counting mode. Then, 128 × 128 pixels images were acquired with pixel dimensions of 400 nm and a pixel dwell time of 25 to 50 µs. A total of 100 scans were collected as image time-stacks using the Zeiss Black ZEN software. A self-written Matlab code (The MathWorks, Natick, MA, USA) was used to analyze the intensity time-stacks data. The Matlab algorithm utilizes the equations from Digman et al. [20] for obtaining the molecular brightness and number as a function of pixel position. We corrected partially for bleaching and minor cell movements using a boxcar-filter with an 8-frame window, applied pixel-wise, as previously described [21,22]. Final brightness values were computed by extrapolating the partial brightness values (i.e., calculated within each 8-frame window) to the earliest time point. Detector saturation, which leads to artefactual reduction in brightness, was avoided by excluding pixels with photon-counting rates exceeding 1 MHz. In order to correct for detector response, we took into account the signal originating from a thin film containing immobilized fluorophores [23]. Region of interest (ROI) were selected manually to exclude large immobile structures (e.g., large protein structures and filaments) and typically contained around 100 pixels. The corresponding brightness values were usually symmetrically distributed around an average value. Ultimately, we obtained the average brightness values from each ROI/cell and normalized them to account for the monomer brightness and the fluorescence probability (pm), which summarizes the detectability of the tag [23]. We calculated the concentration N (in monomer units) by dividing the mean count rate in the ROI by the absolute brightness of the reference monomer, taking into account the pm and the size of the detection volume. The detailed explanation of the calculation is provided in [19].

### 2.7. Statistical Test

In quantitative image analyses, single cells were analyzed separately to address cell-to-cell variance of the parameter under study. If not otherwise stated, the mean ± SEM of individually analyzed cells is displayed. Typically, statistical significance was assessed using Prism (GraphPad Software Inc., San Diego, CA, USA), applying parametric one-way analysis of variance (ANOVA) tests and displayed as follows: **** *p* < 0.0001; *** *p* < 0.001; ** *p* = 0.001–0.01; * *p* = 0.01–0.05.

## 3. Results

### 3.1. Generation and Characterization of Fluorescently Tagged N Protein Constructs

To be able to investigate N protein dynamics using live fluorescence microscopy, we designed fusion proteins, consisting of a N-terminal yellow fluorescent protein (YFP) and the ORF of the PUUV N protein, obtained from cDNAs of in-vitro-infected VeroE6 cells. The N-terminus was chosen for the attachment of the fluorophore over a c-terminal tagging based on previous structural studies reporting a key role of the c-terminal arms in N protein oligomerization [24,25]. First, this construct, henceforward termed YFP-N, was transfected into CHO-K1 cells to provide an initial, unbiased investigation of its overall, intracellular distribution and expression kinetics. Then, we imaged cells at 24 h post transfection (p.t.), thus obtaining a general overview of YFP-N expression pattern at a steady-state among multiple cells. Of note, we found a broad variety of cells with highly diverse intracellular distributions of YFP-N (Figure 1A). Whereas some cells exhibited a punctate expression pattern, others showed fibrillar spikes or arch-like structures extending through most of the cell body (Figure 1A, blue arrows). To ensure that the observed YFP-N distributions are not solely artifacts caused by the N-terminal tagging of the viral protein, we also transfected non-tagged N protein variants into CHO-K1 cells, followed by immunofluorescence staining and imaging, which revealed similar heterogenous structures, including the previously described punctate, fibrillar, and arch-like assemblies (Appendix A).

### 3.2. Dynamics of YFP-N Aggregate Formation

Next, we sought to investigate YFP-N clustering kinetics and dynamics. To this aim, we transfected CHO-K1 cells with YFP-N and monitored YFP-N expression by time-lapse live microscopy for extended periods of time. Interestingly, we observed that YFP-N clustering typically starts in both the perinuclear region and the cell periphery and then extends into other cytoplasmic regions until most of the cell body harbors YFP-N clusters or larger aggregates (Figure 1B, Appendix A). We then imaged several cells for up to 12.5 h p.t. and measured YFP-N aggregation and overall expression by quantitative image analysis using the ImageJ plugin ComDet. Of note, YFP-N expression levels initially increased steadily before reaching a maximum at roughly 10 h p.t. (Figure 1C solid line, Appendix A). In contrast, average cluster-size and fluorescence intensity reached saturation levels already at 5–7 h p.t. (Figure 1C, scatter plots). The respective standard deviations, however, further increased throughout the entire observation period (Appendix A). This finding indicates that YFP-N aggregates with average size (which represent the majority of all YFP-N clusters) are formed at early time points p.t., preceding the protein’s maximum, steady-state expression at the later stages. This notion is further supported by the observation that the cluster number reaches a maximum at 5.5 h p.t. and declines thereafter (Figure 1C, dashed line).

### 3.3. YFP-N Co-Localizes with Vimentin and Actin Fibers

Our initial characterization of YFP-N indicated a preferential cluster formation in the cell periphery, likely lining the plasma membrane. Such distribution is reminiscent of the cortical actin network. Moreover, it was previously reported that different hantaviruses specifically exploit cytoskeleton components, including vimentin and actin [15]. Therefore, to test for an association of YFP-N with the latter two proteins, we performed a specific staining of YFP-N transfected cells using anti-vimentin antibodies and Rhodamine-Phalloidin (actin-staining), respectively (Figure 2A). Of note, YFP-N fibrillar structures often clearly co-localized with actin-spikes, a property that was also found in other mammalian cell lines, including the human lung epithelial line A549 and the bank vole-derived MGLU-2-R culture [17] (Appendix A). The vimentin distribution was more diffuse but nonetheless coincided markedly with YFP-N aggregates (Figure 2A). We next sought to quantitatively evaluate of our fluorescence microscopy images and performed multiple line-plots, statistically analyzing correlation between YFP-N and vimentin or actin (Figure 2B, Appendix A). Lines were selected so to include significant signal in all three channels, and as expected, plots demonstrated a strong association between actin and YFP-N fibers, an observation that corroborated our initial observations. Vimentin, being less distinctly arranged in pin-like structures than actin, still showed a fairly high correlation with YFP-N in multiple cells but with a much higher variability than actin (Figure 2B, Appendix A).

### 3.4. YFP-N Does Not Co-Localize with Microtubules

Vimentin and actin represent two of the three key cytoskeleton components, namely intermediated filaments (vimentin), the actin network, and microtubules. To test for spatial correlation between the latter and YFP-N, we performed immunofluorescence experiments, staining YFP-N transfected cells for tubulin (Figure 3). Both microtubule staining and YFP-N consistently showed tubular/fibrillar structures, which nonetheless showed only minor, if any, co-localization (Figure 3, cell #1 and #2). This held true even in the perinuclear region, where the microtubule organizing center (MTOC) and YFP-N formed a pronounced but independent cluster (Figure 3, cell #2).

### 3.5. YFP-N Puncta Heavily Associated with P-Bodies

All hantaviruses are highly dependent on the cellular transcription machinery. An essential process called cap snatching involves cellular P-bodies (PB), which provide mRNA caps that serve as primers for the replication of the viral genome by the hantavirus RNA dependent RNA polymerase (RdRp) [6,8]. N proteins and RdRp have been proposed to be involved in that process. Here, we sought to test whether YFP-N, in absence of other viral components, translocates to PBs or if this process requires vRNAs, RdRp, or viral glycoproteins. YFP-N-transfected cells were stained for PBs using antibodies against the PB-resident protein Dcpa1. A noticeable correlation was found between PBs and larger YFP-N aggregates but even more pronounced, with smaller YFP-N puncta (Figure 3B and Appendix A). PBs seemed to align with fibrillar YFP-N structures, whereas punctate structures showed high degrees of co-localization.

### 3.6. Unsupervised, Quantitative Image Analysis Corroborates YFP-N Co-Localization with PBs, Actin, and Vimentin

Our fluorescence microscopy images indicated a marked association of YFP-N with actin, vimentin, and PBs but not with microtubules. Initially, we performed some image quantification of actin-stained cells by line-plot analyses. Now, in order to avoid any observer bias in our quantitative analysis, we next performed automated image segmentation and quantification (Appendix A) using the cell profiler software package [18]. Pearson correlations were obtained for all above shown microscopy experiments as well as an additional positive control staining of YFP-N transfected cells with a commercial N-protein antibody. This quantitative data analysis strongly supports our above-described observations, demonstrating that YFP-N significantly co-localizes with vimentin, PBs, and actin but not with tubulin (Figure 4).

### 3.7. YFP-N Co-Localizes with Gc and Gn

We previously generated fluorescently tagged versions of both hantavirus glycoproteins, Gc and Gn [21]. Our recent work unequivocally demonstrated that upon co-expression these two proteins are highly enriched in the perinuclear region, whereas separately expressed, Gc and Gn show different localization patterns. We thus surmised that Gc and Gn are interacting in the endoplasmic reticulum, thus mutually promoting trafficking to the Golgi/ERGIC [21]. Now, we were interested to assess whether mTurquoise-N, a cyan-fluorescent version of YFP-N, would equally co-localize with the mCherry-Gc and YFP-Gn. In fact, we observed strong spatial correlation in fluorescence microscopy images and found significant co-localization between all three proteins by automated image analysis. Noteworthily though, co-localization was significantly more pronounced between Gc and Gn than between mTurquoise-N and either glycoprotein (Figure 5, Appendix A).

### 3.8. N protein Multimerization Is Dose-Dependent and Independent of Hantavirus Glycoproteins

Hantavirus N protein has previously been reported to form dimers, trimers, and high-order multimers [11,26,27,28]. We employed Number and Brightness analysis to study YFP-N self-assembly directly in CHO-K1 cells. This fluorescence fluctuation spectroscopy technique determines the fluorescence intensity (molecular brightness) of individual protein complexes (molecular brightness), thus effectively assessing the oligomerization state of the protein under investigation with a single-cell resolution. We previously utilized this method to assess oligomerization of both Gc and Gn [19,21] and used it here to further characterize YFP-N. We also sought to test whether interactions between the hantavirus structural proteins affect oligomer assembly, thus monitoring YFP-N oligomerization in presence and absence of Gc and Gn (both tagged with the fluorescent protein mCherry2). We focused on cells with relatively low expression levels of YFP-N to capture early oligomerization events and therefore the dynamics of the self-assembly process. Large immobile protein structures cannot be analyzed via N&B and were, therefore, excluded during ROI selection. Expectedly, when YFP-N was transfected alone, we found a marked correlation between overall YFP-N expression levels and YFP-N oligomerization, indicative of a high self-affinity of the viral protein (Figure 6A,B green dots and curves). YFP-N formed dimers, trimers, and high-order multimers consisting of up to ten individual proteins. Noteworthily, co-expression of either hantavirus glycoprotein did not significantly alter YFP-N self-assembly (Figure 6A,B red dots and curves), which showed a similar multimerization behavior in absence and presence of either Gc or Gn.

## 4. Discussion

In this study, we investigated fluorescently labelled variants of the Puumalavirus nucleoprotein (N). Two chimeric constructs, YFP-N and mTurquoise-N, were generated and analyzed in the rodent cell line CHO-K1. Our study focused on CHO-K1 cells because the cell biology of this widely used hamster cell line is extremely well described and because they enable reliable, efficient, and highly reproducible transfection and live-cell microscopy. CHO-K1 cells per se are not permissive for hantavirus infections; however, rodents are the natural vector of hantaviruses, and several hamster models closely recapitulate hantavirus pathobiology and disease [29,30].

We assessed the intracellular time-resolved trafficking and localization of our N protein constructs as well as their co-localization with selected cellular proteins. In this context, we specifically focused on the cytoskeleton proteins-tubulin (microtubular network), vimentin (intermediate filaments), and actin (actin network), thus looking at all branches of this crucial morphological cellular structure. We also assessed co-localization with P-bodies, which are critically involved in hantavirus replication cycles [8]. Finally, we performed co-transfection experiments with the viral glycoproteins Gc and Gn, both fluorescently labelled, observing significant co-localization of all three viral proteins, particularly in the perinuclear region. To be able to draw reliable objective conclusions from the microscopy study, we conducted extensive automated image analyzes. We have focused on image segmentation using the cell profiler platform, developed by the Broad Institute, since it enables rapid, unbiased cell identification and image quantification once a viable analysis pipeline was generated. We report pixel-by-pixel Pearson correlation, a measure of overall co-localization between two markers (fluorescent proteins and immunofluorescence staining), assessed on a single-cell level, following automated image segmentation. Finally, we conducted oligomerization studies based on live, single-cell experiments using the Number and Brightness technique [31].

In our initial experiments, we observed that YFP-N, early after transfection, begins to form small, punctate clusters, which rapidly coalesce first into pin-like structures of only a few micrometers length and ultimately into larger fibrillar aggregates that extend throughout large parts of the cytoplasm (Figure 1). To the best of our knowledge, this is the first comprehensive description of elongated, fibrillar structures formed by Puumalavirus N protein. A recent study showed similar N protein clusters, which were, however, not further investigated [32]. Of note, structures reminiscent of these YFP-N aggregates have been recently observed in persistent, long-term Tula virus [33] and Hantaan virus infections [15,16] but were not found during PUUV infections [16]. However, cluster formation is highly cell-type and context dependent [16], and it seems likely that oligomerization-clustering properties of N protein are shared across different old-world hantaviruses, in particular between phylogenetically closely related viruses, such as Tula virus and PUUV [34]. Our study unequivocally demonstrates that PUUV N forms fibrillar structures in absence of fluorescent tags (Appendix A) as well as in multiple mammalian cells lines (Appendix A), including bank vole and human lung cells lines, thus indicating the large-scale clustering is an inherent and target cell-independent property of this viral protein. YFP-N clustering typically starts and proceeds essentially in the perinuclear region and, importantly, also in the cellular periphery. Perinuclear accumulation of viral proteins is considered a hallmark of hantavirus infections, reported for both new- and old-world hantavirus species [35,36]. Peripheral localization and, in particular plasma membrane association, however, is rather controversial, with only limited evidence for an involvement of the plasma membrane in hantavirus post-entry processes [37,38]. Our earlier studies indicated a significant albeit low exposure of PUUV glycoproteins at the cell surface [21], suggesting that this process is not restricted to new-world hantavirus as proposed previously [38,39].

General membrane association of N has been described early and is mediated through electrostatic interactions, likely by residues at the C-terminus of the protein [35]. We however surmise that the plasma membrane association we have observed in our live-cell experiments is indicative of interactions with the cortical actin rather than direct interactions with the bilayer or surface exposed proteins. This hypothesis is supported by our immunofluorescence staining and image analysis results, showing marked co-localization between YFP-N and actin (Figure 2, Figure 4). Specifically, filamentous YFP-N structures exhibit a remarkable similarity to and co-localization with actin filaments (Figure 2B). We also found significant co-localization of YFP-N with vimentin but not with tubulin (Figure 2, Figure 3, Figure 4), indicative of specific interactions with either actin and vimentin directly or with other cellular proteins being associated with the respective cytoskeletal structures. Previous reports have highlighted the crucial function of the cytoskeleton during hantavirus infections [5,9,15,40]; however, compelling evidence for direct interactions between individual hantavirus proteins and cytoskeleton proteins is scarce. Our data now strongly suggest that there could be a direct contact of N with actin and intermediate filaments, even in absence of any other viral components.

We furthermore observed significant co-localization between YFP-N and the P-body marker DCPa1 (Figure 3, Figure 4). P-bodies are believed to be responsible for storage and degradation of cellular RNA and have been implicated in hantavirus replication mechanisms by providing primers for viral mRNA synthesis in a process termed cap-snatching. Mir and colleagues have reported N P-body association for Sin Nombre virus, a new-world hantavirus [8,41]; however, to date, no old-world hantavirus has been investigated for such interactions. Of note, actin-interactions of N have been proposed to be involved in the trafficking of N to their P-body destination [15,40], underlining the key importance of actin for a successful hantavirus infection and replication.

We also performed co-expression experiments with PUUV Gc and Gn, all fluorescently tagged. A strong co-localization was found between Gc or Gn with N albeit less pronounced than co-localization between both glycoproteins. This is not surprising given that Gc and Gn are membrane proteins that are derived from a common precursor, whereas N is synthesized independently as a soluble, cytoplasmic protein. Nonetheless, co-localization between YFP-N and Gc/Gn could suggest a direct interaction between these viral proteins. However, our experimental approach cannot distinguish between physical interactions and co-enrichment in the same microenvironment (or cellular compartment).

Finally, we conducted N&B experiments to study self-assembly of YFP-N clusters in presence and absence of Gc and Gn. We found a strong correlation between YFP-N oligomerization and overall expression levels when YFP-N was transfected alone, suggesting a concentration-dependent cluster formation. We found monomers, dimers, trimers, and larger multimers consisting of around 10 individual proteins. This is in agreement with previous publications, reporting evidence from mostly biochemical approaches, for a broad spectrum of N protein oligo- and multimers [11,26,27,28]. Of note, we are the first to investigate N protein oligomerization in live-cell microscopy experiments, thus providing direct evidence for the formation of high-order N protein complexes *in cellulo*. Importantly, the concentration dependency was not majorly altered in presence of either Gc or Gn, indicating that the viral glycoproteins do not interfere with YFP-N clustering. Future studies will have to investigate whether viral RNAs or other viral proteins contribute to N protein clustering, thus inducing the formation of even larger macromolecular complexes and ultimately complete viral particles.

## Figures and Tables

**Figure 1 viruses-14-00457-f001:**
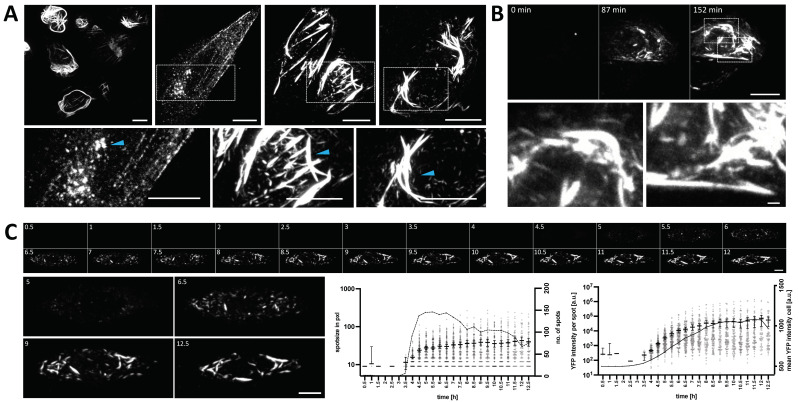
Heterogeneity and dynamics of YFP-N expression patterns. (**A**) CHO-K1 cells were transfected with YFP-N for 24 h and observed via confocal spinning disk microscopy. Transfected cells show highly diverse and heterogenous distributions of YFP-N throughout most of the cell body. The image on the upper left shows an overview of multiple cells. Individual examples are shown on the right and magnifications of the boxed areas are displayed at the bottom of the panel. Blue arrows highlight specific protein aggregation states (punctate, spikes, and arch-like tubules). (**B**) The dynamics of YFP-N assemblies were observed by live time-lapse microscopy over several hours post transfection. Three time points are shown in the upper panel. The two boxed areas at 152 min are shown magnified in the lower panel (Scale bar: 1 μm). (**C**) Representative cell (top panel), repeatedly imaged for 12.5 h post transfection. Each individual micrograph shows the same cell, imaged in 30-min intervals. Another panel (bottom left) displays magnifications of four selected time points as indicated on the images. The graph on the on the bottom right shows size and fluorescence intensity of single protein aggregates plotted over time (grey dots, see Appendix A for additional examples). Protein aggregates were identified by automated image analysis using the ImageJ plugin ComDet. Bars show the mean with 95% confidence interval. The solid lines indicate the overall mean YFP intensities of the entire cell, whereas the dashed line shows the overall number of detected spots (both plotted on right Y-axes). All micrographs show maximum intensity projections of z-stacks. If not otherwise stated scale bars: 10 μm.

**Figure 2 viruses-14-00457-f002:**
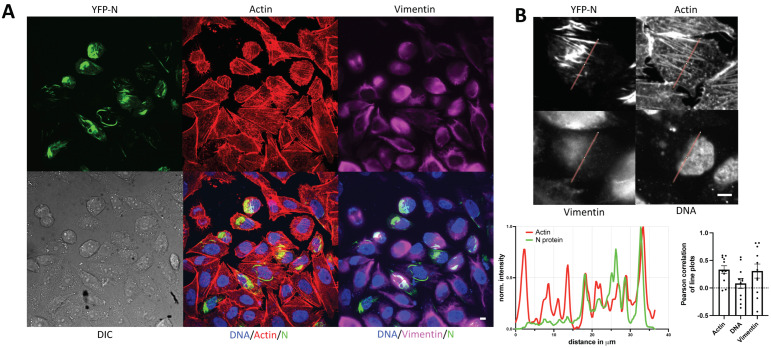
YFP-N co-localizes with cellular actin and vimentin. (**A**) CHO-K1 cells were transfected with YFP-N and stained 24–48 h p.t. for actin using Rhodamine-Phalloidin and vimentin by immunofluorescence. (**B**) Line-plot analysis of individual cells reveals marked co-localization of YFP-N and actin filaments as well as vimentin. The panel shows one representative cell and the corresponding line plots from all fluorescence channels after normalization. The bar chart on the lower right shows a quantitative analysis of line plots from multiple cells (*n* > 10). Pair-wise Pearson correlations of line plots were calculated for YFP-N with the cellular markers as stated in the bar chart. Bars show mean with SEM. Additional examples can be found in the Appendix A. All images show maximum intensity projections of z-stacks. Scale bars: 10 μm.

**Figure 3 viruses-14-00457-f003:**
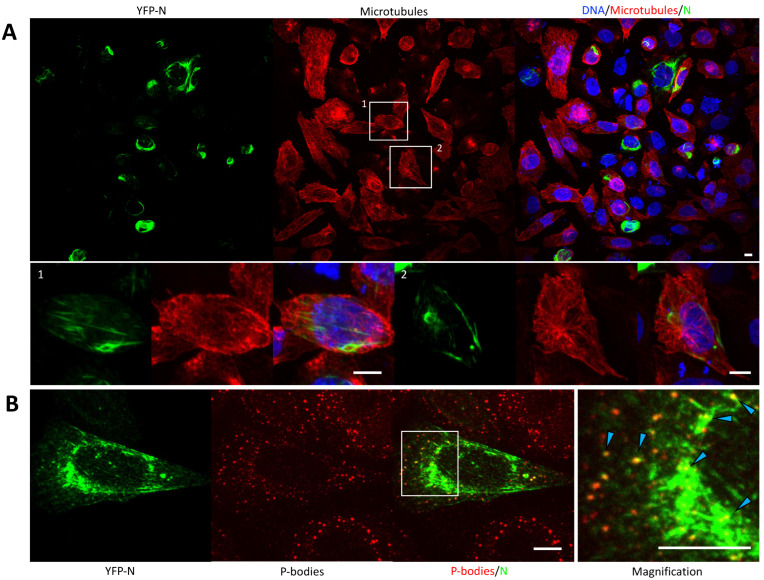
YFP-N co-localizes with P-bodies. (**A**) CHO-K1 cells were transfected with YFP-N and stained 24–48 h p.t. for microtubules by immunofluorescence. The two boxed cells are shown magnified in the lower panel. (**B**) CHO-K1 cells were transfected with YFP-N and stained for P-bodies by immunofluorescence using anti-Dcpa1 antibodies. A magnification of the boxed area is shown on the right. Punctate patterns with clear co-localization of P-bodies and YFP-N are highlighted with arrows. All images show maximum intensity projections of z-stacks. Scale bars: 10 μm. Additional examples can be found in the Appendix A.

**Figure 4 viruses-14-00457-f004:**
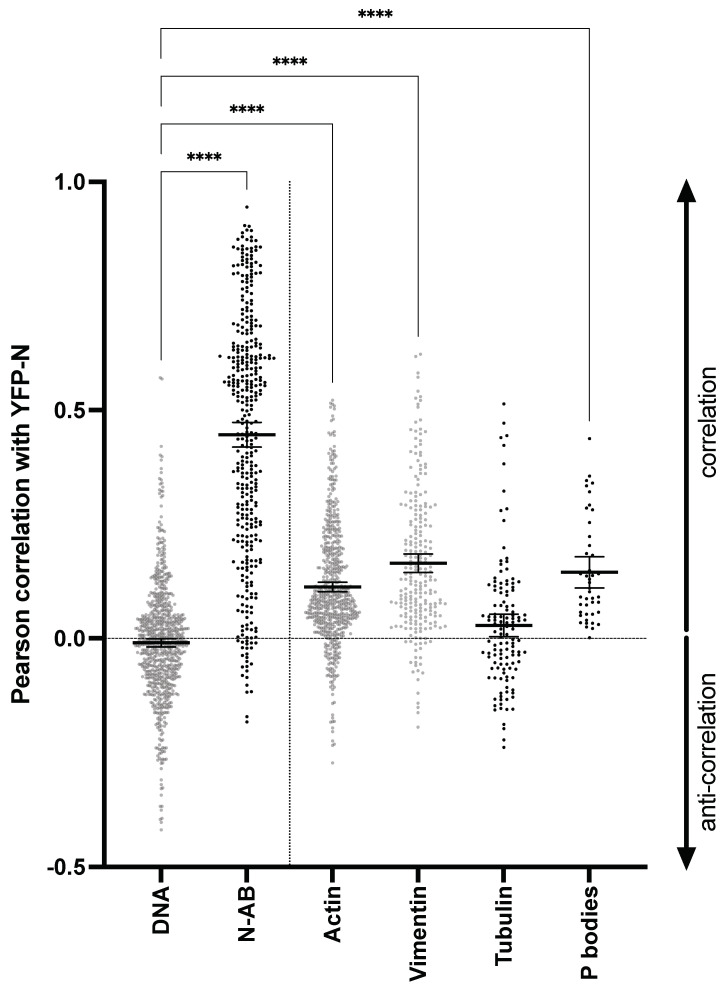
Quantitative automated image analysis reveals association of YFP-N with actin filaments, vimentin, and P-bodies but not tubulin. Micrographs as shown in Figure 1, Figure 2 and Figure 3 were subjected to automated image analysis using cell profiler. Image segmentation examples are shown in the Appendix A and described in detail in the material and methods section. Bars show Pearson correlation of YFP-N with the markers as stated. Dots show individual cells (*n* > 20). DNA staining is utilized as a negative control staining assuming negligible YFP-N expression in the nucleus. N protein antibody staining (N-AB) was employed as a positive control. All bars show mean with SEM. Significance was tested using a one-way analysis of variance (ANOVA), **** *p* ≤ 0.0001.

**Figure 5 viruses-14-00457-f005:**
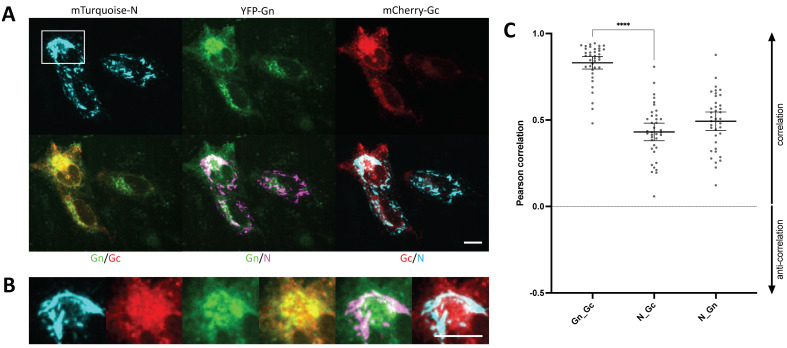
mTurquoise-N co-localizes with fluorescently tagged Gc and Gn. (**A**) CHO-K1 cells were transfected with mTurquoise-N (shown in cyan), YFP-Gn (green), and mCherry-Gc (red) and imaged 24–48 h p.t. (**B**) Magnification of the boxed region shown in (**A**). Co-localization in overlay images appears orange (green and red), white (green and magenta, or red and cyan). Scale bars: 10 μm. Additional examples can be found in the Appendix A. (**C**) Pearson correlation between proteins as stated in the bar chart were assessed by automated image analysis as described in Figure 4. All bars show mean with SEM (*n* = 38). Significance was tested using a one-way analysis of variance (ANOVA) **** *p* ≤ 0.0001.

**Figure 6 viruses-14-00457-f006:**
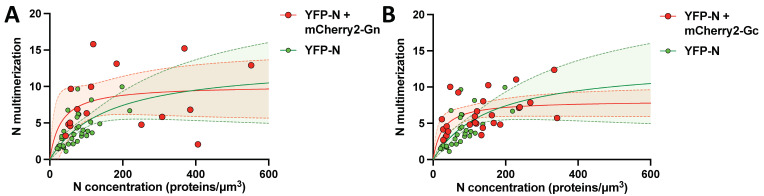
YFP-N forms high-order multimers in presence and absence of Gc and Gn. YFP-N multimerization as a function of total protein concentration (in monomer units) in the absence and presence of (**A**) mCherry2-Gn or (**B**) mCherry2-Gc expressed in CHO-K1 cells. Each point in the graph represents the average NP multimerization and concentration within a ROI in one CHO-K1 cell. Solid lines show non-linear fits to a binding kinetic model (Y = Bmax × X/(Kd + X)) and dashed lines indicate 95% confidence intervals. The fits are of relatively low statistical significance and should be viewed as a qualitative guide to the eye.

**Table 1 viruses-14-00457-t001:** Reagents for fluorescence and immunofluorescence staining.

**Primary Antibodies**	**Manufacturer**
Hantavirus (Puumala) N protein antibody	Fitzgerald, UK (cat. 10R-2502)
Anti-Vimentin antibody	Abcam, Cambridge, UK (cat. ab45939)
Anti-Dcp1a antibody	Abcam, Cambridge, UK (cat. ab57654)
Anti-tubulin (clone B512)	Sigma-Aldrich, Munich, Germany (cat. T5168)
Fluorescently conjugated phalloidin	Thermo Fisher Scientific, Waltham, MA, USA (cat. R415)
**Secondary Antibodies**	**Manufacturer**
AlexaFluor 488 goat anti rabbit IgG (H + L)	Invitrogen, Carlsbad, CA, USA (cat. A-11008)
AlexaFluor 488 goat anti mouse IgG (H + L)	Abcam, Cambridge, UK (cat. ab150117)
AlexaFluor 568 goat anti mouse IgG (H + L)	Invitrogen, Carlsbad, CA, USA (cat. A-11004)
Alexa fluor 594 goat anti rabbit IgG (H + L)	Invitrogen, Carlsbad, CA, USA (cat. A-11012)
Alexa fluor 647 goat anti mouse IgG (H + L)	Invitrogen, Carlsbad, CA, USA (cat. A-21235)
Alexa fluor 647 goat anti rabbit IgG (H + L)	Abcam, Cambridge, UK (cat. ab150087)

## Data Availability

Appendix A are provided. Raw data are available upon request.

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
