# Peer review of "Characterization of Hantavirus N Protein Intracellular Dynamics and Localization"

_viruses, 2022, doi:10.3390/v14030457_

Round 1

Reviewer 1 Report

The work of Welke et al. represents a co-localization study of PUUV nucleocapsid protein with cellular and viral proteins. They observed a partial co-localization with vimentin, actin and P-bodies and a multimerization of nucleocapsid proteins. The microscopy methods are of high quality. However, as mentioned by the authors, co-localization and multimerization was already examined and shown for hantaviral nucleocapsid proteins by several other studies. Therefore, the presented work lacks novelty. The “co-localization” with P-bodies is very weak and not convincing (especially in the supplemental data). It is not comparable with New World viruses and has to be confirmed by further studies, e.g. untagged protein and infection studies, to demonstrate the relevance. Different cell types were used, CHO cells are not target cells of hantaviruses, the host rodent cells shown in the supplemental data are of special interest, but were excluded from the main manuscript. In addition, some minor points weaken the quality of the manuscript.

Line 36: order Bunyavirales, family Hantaviridae (see hantaviral taxonomy, ICTV)

N-YFP: I would prefer YPF-N (as labeled in the supplementary figure 2A), because it is tagged at the N-terminus

Line 150: 105

Line 425: video? Not present in the submitted files.

Supplement: Explain the MGN cell line. Species, cell type?

Reviewer 2 Report

Please find my comments attached.

Round 2

Reviewer 1 Report

The overall criticism is that CHO cells are used without showing the more relevant cell types in the main text and that the results are not confirmed in infection studies. These points are not adequately addressed in the revised version.

In addition, the publications, which are cited to justify the use of CHO cells, do not use these cells for localization or post-entry studies and hamsters are not animal models for Old World hantaviruses.

The data from infection studies (cell type ?) presented in the point-by-point response are interesting and have to be included in this manuscript to demonstrate the relevance of transfection experiments. Furthermore, the localization of NPUUV was already analysed and quantified in transfected BHK cells by Ravkov et al. in 2001. They observed a filamentous NPUUV in 5% of cells. In the presented manuscript it is not clear how many CHO cells show this pattern. In a newer study by Haegel et al. a filamentous pattern was observed in single cells of a podocyte cell culture. All these facts are not mentioned in the manuscript. A carefully prepared manuscript showing the results from host animal cells and infection studies in the main text together with an appropriate discussion of the results with regard to the existing literature would probably be suitable for publication.